# Enhanced Light Absorption and Efficient Carrier Collection in MoS_2_ Monolayers on Au Nanopillars

**DOI:** 10.3390/nano12091567

**Published:** 2022-05-05

**Authors:** Jungeun Song, Soyeong Kwon, Hyunjeong Jeong, Hyeji Choi, Anh Thi Nguyen, Ha Kyung Park, Hyeong-Ho Park, William Jo, Sang Wook Lee, Dong-Wook Kim

**Affiliations:** 1Department of Physics, Ewha Womans University, Seoul 03760, Korea; sje10056996@gmail.com (J.S.); kwonsso91@gmail.com (S.K.); hjjeongssi@gmail.com (H.J.); hjchoi1214@gmail.com (H.C.); nthianh111@gmail.com (A.T.N.); hakyungpark@ewhain.net (H.K.P.); wmjo@ewha.ac.kr (W.J.); leesw@ewha.ac.kr (S.W.L.); 2Nanodevice Laboratory, Korea Advanced Nano Fab Center, Suwon 16229, Korea; hyeongho.park@kanc.re.kr

**Keywords:** MoS_2_, surface plasmon, photoluminescence, electron transfer, surface photovoltage

## Abstract

We fabricated hybrid nanostructures consisting of MoS_2_ monolayers and Au nanopillar (Au-NP) arrays. The surface morphology and Raman spectra showed that the MoS_2_ flakes transferred onto the Au-NPs were very flat and nonstrained. The Raman and photoluminescence intensities of MoS_2_/Au-NP were 3- and 20-fold larger than those of MoS_2_ flakes on a flat Au thin film, respectively. The finite-difference time-domain calculations showed that the Au-NPs significantly concentrated the incident light near their surfaces, leading to broadband absorption enhancement in the MoS_2_ flakes. Compared with a flat Au thin film, the Au-NPs enabled a 6-fold increase in the absorption in the MoS_2_ monolayer at a wavelength of 615 nm. The contact potential difference mapping showed that the electric potential at the MoS_2_/Au contact region was higher than that of the suspended MoS_2_ region by 85 mV. Such potential modulation enabled the Au-NPs to efficiently collect photogenerated electrons from the MoS_2_ flakes, as revealed by the uniform positive surface photovoltage signals throughout the MoS_2_ surface.

## 1. Introduction

Two-dimensional (2D) layered transition metal dichalcogenides (TMDs) are emerging as candidates for novel electronic and optoelectronic devices due to their fascinating physical properties, including their large exciton binding energy, high electron mobility, and superior mechanical flexibility [1,2,3,4,5,6,7,8,9,10,11,12,13,14,15,16,17,18,19,20]. Metal electrodes are indispensable parts of all devices based on TMDs and conventional semiconductor (SC) materials [10,11,12,13]. The electrodes can inject and charge carriers into and collect charge carriers from active SC materials to operate the devices. Deposition and patterning processes of metal layers on or under ultrathin TMD layers require special care, since the physical properties of 2D TMD materials are quite distinct from those of 3D bulk counterparts [1,2,3,4,5,6,7,8,9,10,11,12,13]. Intensive efforts based on experimental and theoretical research activities have led to proposals of useful approaches for fabrication and electrical characterization of TMD/metal contacts [10,11,12,13]. However, a universal route to achieve desirable metal contacts for TMD-based devices is still lacking [11,13].

There have been many successful demonstrations of high-performance optoelectronic devices that use nanopatterned metal layers [1,2,3,4,5,21,22]. Patterned metal electrodes, rather than flat electrodes, can improve the light-matter interactions in SCs with the help of surface plasmons (SPs). In metal nanostructures, two types of SPs can be excited: localized surface plasmons (LSPs) and propagating surface plasmon polaritons (SPPs) [23]. LSP effects strongly confine incident light near metal nanostructures, especially at resonance wavelengths [1,3,4,5,6,7]. Propagating SPP excitation at the dielectric/metal interface leads to the generation of evanescent waves perpendicular to the interface, and hence, an intense electric field appears on the metal surface [2,8,21,22]. Both LSP and SPP can boost the optical absorption in SC layers on and under metal nanostructures [1,2,3,4,5,6,7,8,9,21,22]. Bottom-up and top-down techniques can be used to fabricate metal nanostructures in SC/metal hybrid systems. The former enables low-cost high-throughput fabrication but cannot produce well-ordered patterns [1,6,9]. In contrast, the latter can produce regular periodic nanostructures with high precision. Electron-beam lithography is one of the most popular top-down techniques due to its high-resolution capability and compatibility with subsequent nanofabrication processes [1,2,5,7,8]. Although the absorption coefficients of TMD materials are much larger than those of conventional SCs, the optical absorption in TMDs is limited due to their extremely small physical thickness. Thus, integration of TMDs with metal nanostructures has been attempted to overcome the limited absorption and tune the spectral response of atomically thin TMDs [1,2,3,4,5,6,7,8,9,14,15,16]. The plasmonic effects in TMD/metal nanosystems improve the performance of various TMD-based devices, including the photodetectors [3], photocatalysis devices [4], and bio-sensors [5].

The optical benefits of TMD/metal nanostructures have been reported in numerous works, as discussed above. However, few studies have directly investigated the charge collection capability of metal nanostructures in TMD/metal hybrid systems. The electrical transport along the lateral direction in the TMD layers as well as that along the vertical direction at the TMD/metal interface should be considered to examine the carrier collection behaviors of the TMD/metal hybrid structures. Moreover, most TMD samples consist of micron-size flakes. Therefore, the electrical characterization of TMD-based nanostructures requires tools with a high spatial resolution. Kelvin probe force microscopy (KPFM) can measure the contact potential difference (CPD) of the sample and provide surface charge distribution with a spatial resolution of a few tens of nm [8,9,13]. The CPD measurements have been used to reveal the band bending and band offset in SC heterojunctions, which can cause separation and drift of photogenerated charges in SCs [8,9,24]. Light-induced CPD change is referred to as the surface photovoltage (SPV), and KPFM-based SPV measurements can be used to visualize the spatial distribution of photogenerated charge carriers in nanostructures [25]. Thus, the KPFM characterizations will help us to study the behaviors of the charge carriers in TMD/metal hybrid nanostructures.

In this work, we fabricated MoS_2_/Au hybrid nanostructures and investigated their physical characteristics. SiO_2_ nanopillar arrays were fabricated using electron-beam lithography, and then Au thin films were evaporated onto them under high vacuum. Such patterning of SiO_2_ and subsequent coating of Au thin films provide a simple means of producing periodic Au nanopillars (Au-NPs). The optical spectra and calculations of the hybrid systems showed how the plasmonic Au-NPs influenced the light-matter interaction in the MoS_2_ monolayers transferred onto the Au-NPs. SPV characterizations of the MoS_2_/Au-NPs allowed us to visualize the spatial distributions of photogenerated charge carriers under light illumination.

## 2. Materials and Methods

Figure 1a,b show a cross-sectional schematic diagram and a top-view optical microscope image of the Au-NP array. First, 50-nm-high SiO_2_ NP arrays (period: 500 nm, top diameter: 250 nm, bottom diameter: 300 nm, and height: 50 nm) were fabricated on SiO_2_ (300 nm)/Si wafers using electron-beam lithography and dry etching techniques, as reported in our earlier work [25]. 10-nm-thick Ti adhesion layers were deposited on the SiO_2_ NP arrays, and 100-nm-thick Au thin films were deposited on those layers using electron-beam evaporation. Periodic arrays of plasmonic Au-NPs can be produced through simple evaporation of Au thin films on SiO_2_-NPs. The nanofabrication processes of SiO_2_ are well established and widely used. Thus, the use of SiO_2_-nanopattern templates provides a simple approach to obtain Au nanostructures without etching of Au thin films [21,22].

The area of the Au-NP array was 500 × 500 μm^2,^ and there was a flat region around the Au-NP array, as shown in Figure 1b. This flat region coated with the Au/Ti thin films is referred to as Au-F, and it can be used for comparative characterizations. The low- and high-magnification scanning electron microscope (SEM) images in Figure 1c show regular arrays of NPs and confirm the thickness of the Au (100 nm)/Ti (10 nm) films, respectively. The highly directional evaporation in our high vacuum chamber (base pressure: 10^−7^ Torr) limits deposition at the NP sidewalls. Consequently, the surface morphology of Au-NPs is likely identical to that of the underlying SiO_2_ NPs, as illustrated in Figure 1a. The high-magnification cross-sectional SEM image in Figure 1c confirms this expectation: the top and bottom diameters of the Au-NPs (250/300 nm) were almost identical to those of the SiO_2_ NPs. The SEM images also show that the 100-nm-thick Au thin films formed continuous layers over the underlying SiO_2_ NPs. The thickness of the Au thin film was 2 times larger than the penetration depth of Au at a wavelength (*λ*) of 600 nm [23]. Thus, the underlying SiO_2_/Si substrates did not affect the optical characteristics of Au-NPs. Monolayer MoS_2_ flakes grown by chemical vapor deposition were transferred onto the Au-NPs using a polymer-based wet transfer technique. Detailed procedures have been reported in our earlier publications [8,9]. The polymer was removed in a critical point dryer using acetone, and hence, the surface of the transferred MoS_2_ flakes on Au-NPs remained very flat because the surface tension-induced strain in the suspended MoS_2_ region was relieved. Optical reflectance from a selected area of several μm^2^ was measured using a homemade setup consisting of an optical microscope (LV100, Nikon Instruments, Seoul, Korea), a spectrometer (Maya 2000 Pro, Ocean Insights, Rostock, Germany), and a white LED (Solis-3C, Thorlabs, Newton, NJ, USA) [25,26]. The measured optical spectra were compared with the calculated spectra obtained from finite-difference time-domain (FDTD) simulations (Lumerical FDTD Solutions, Vancouver, BC, Canada). The optical properties of the MoS_2_ flakes were characterized using a micro-Raman and photoluminescence (PL) spectroscopy setup (XperRam C, Nanobase, Seoul, Korea) with a 532-nm-wavelength laser as a light source.

The surface morphology and CPD of the samples were simultaneously measured using atomic force microscopy (AFM) (NX10, Park Systems, Suwon, Korea) with Pt/Ir-coated Si cantilevers (NGS01/Pt, NT-MDT, Apeldoorn, The Netherlands). All the measurements were carried out in a N_2_-purged glove box to avoid artefacts caused by ambient gas adsorption [8,9,13]. The CPD data were obtained using the amplitude-modulation KPFM mode, where the amplitude and frequency of the AC voltage were 2 V and 17 kHz, respectively. Dark-state CPD data were obtained from the samples after they were stored in the dark for at least 1 h. Light-induced CPD changes were measured while irradiating the samples using a fiber-coupled 660-nm-wavelength light emitting diode (M660L4, Thorlabs, Newton, NJ, USA) and a focusing lens.

## 3. Results and Discussions

Figure 2a,b show a schematic diagram and an optical microscope image of the MoS_2_ monolayer flakes transferred onto Au-F and Au-NP, respectively. Triangular-shaped flakes are clearly visible on the Au-F and Au-NP surfaces (Figure 2b). Raman spectra of the MoS_2_ flakes on Au-F and Au-NP are shown in Figure 2c. The spacing between the in-plane (E_2g_) and out-of-plane (A_1g_) phonon mode peaks is 20.9 cm^−1^, which confirms the preparation of monolayer MoS_2_ flakes [27]. We can note that the Raman peak spacing is identical in both MoS_2_/Au-F and MoS_2_/Au-NP. The Raman peak positions are sensitive to the strain states of the flakes [17,18,19]. The Raman spectra show that the flakes on Au-NPs are nonstrained, similar to those on Au-F. Additionally, the Raman intensity of the MoS_2_ monolayers on the Au-NPs was 3 times higher than the intensity of those on the Au-F. The peak enhancement suggests that Au-NPs could improve the optical absorption of the flakes.

Figure 3a,b show the measured and FDTD-calculated reflectance (*R*) spectra of Au-F and MoS_2_/Au-F, respectively. Figure 3c,d show the measured and calculated *R* spectra of Au-NP and MoS_2_/Au-NP, respectively. Optical reflectance from a selected area of several μm^2^ was measured using a homemade microscope-based setup [25,26]. When *λ* was smaller than 600 nm, the measured *R* values of Au-F and Au-NP decreased due to the interband transition of Au [23]. Transfer of the MoS_2_ monolayers reduced the *R* of both Au-F and Au-NP, presumably due to the optical absorption in the MoS_2_ flakes. The reduction in *R* was more notable in the MoS_2_/Au-NP than in the MoS_2_/Au-F, which may indicate improved absorption in the MoS_2_ flakes on Au-NP. Near *λ* = 600 nm, a broad dip appeared in the measured *R* of Au-NPs, and the calculated *R* exhibited a much sharper dip. This reflection dip likely originated from plasmonic effects on the Au-NP surface, since such a dip was not observed in the bare SiO_2_ NP array [25]. Periodic Au nanostructures can excite propagating SPP along the surface and produce an evanescent field in a direction perpendicular to the surface [8,9,21,22,23]. The spatial period determines the photon-SPP coupling wavelength based on the SPP dispersion relation [23]. The Au nanogratings with a 500-nm period enable SPP excitation at *λ* of 500~600 nm [8,9]. In the spectrum of MoS_2_/Au-NP, the dip shifted again to *λ* ~ 600 nm due to the large refractive index of MoS_2_. This reflectance dip is close to the A and B exciton resonance wavelengths of MoS_2_, as marked by the gray dashed lines in Figure 3c,d [28]. Therefore, the SPP-photon coupling in Au-NPs can create a large number of excitons in the MoS_2_ monolayers, resulting in pronounced dips near the exciton resonance wavelengths. The discrepancies between the experimental and simulated data can be attributed to the imperfect geometric configuration and rough surface of the fabricated Au-NPs.

Figure 4a shows the PL spectra of MoS_2_/Au-F and MoS_2_/Au-NP. The PL intensity of MoS_2_/Au-NP is 20 times larger than that of MoS_2_/Au-F. It should be noted that electron transfer from MoS_2_ to Au suppresses the radiative recombination of photogenerated excitons, resulting in PL quenching of the MoS_2_ flakes on the flat Au thin film [8,9]. Consequently, the PL intensity of MoS_2_/Au-F is smaller than that of MoS_2_ flakes on SiO_2_/Si substrates (Appendix A). In the MoS_2_/Au-NP, the area fraction of the suspended MoS_2_ region was 4/5, and hence, only 1/5 of the area of the MoS_2_ flakes remained in contact with the Au-NP top surface. This reduced contact area can prevent PL quenching; however, the area difference alone could not have led to the observed 20-fold increase in the PL intensity. Figure 4b shows the spectra of the FDTD-calculated optical absorption in the MoS_2_ monolayers (*A*_MoS2_) on Au-NPs and Au-F. *A*_MoS2_ could not be measured from the experiments since the Au layers and the Si substrates also absorb incident light. Thus, *A*_MoS2_ was estimated from the FDTD simulations. Au-NPs significantly increase *A*_MoS2_ in a broad wavelength range. In particular, *A*_MoS2_ of the MoS_2_/Au-NP at a *λ* of 532 nm (the PL and Raman excitation wavelength) is 2 times larger than that of MoS_2_/Au-F. The enhanced excitation should increase the Raman and PL intensities of MoS_2_/Au-NP [2,14]. It should also be noted that the enhanced absorption near the A and B exciton resonance wavelengths of the MoS_2_ monolayer was pronounced, as expected from the *R* spectra. The maximum *A*_MoS2_ of the MoS_2_/Au-NP reached 60% at a *λ* of 615 nm, which was 6 times larger than the *A*_MoS2_ of the MoS_2_/Au-F.

Figure 5a–c show the cross-sectional distributions of the FDTD-calculated electric field (*E*) in our samples under light illumination at *λ* = 530 nm, 615 nm, and 660 nm. In these calculations, the MoS_2_ monolayers were assumed to have flat surfaces. The surface morphology data of the samples will be presented later. The calculated results clearly show that Au-NPs concentrate the incident light. The NP-concentrated light at *λ* = 615 nm and 660 nm (the B and A exciton resonance wavelengths of the MoS_2_ monolayer) is more pronounced than that at *λ* = 530 nm (the Raman and PL excitation wavelengths). In fact, the *A*_MoS2_ of the MoS_2_ flake is proportional to |*E*|^2^ [29]. The comparison of the |*E*|^2^ distributions (Figure 5a–c) explains the wavelength dependence of the *A*_MoS2_ of MoS_2_/Au-F and MoS_2_/Au-NP (Figure 4b). The PL and Raman intensity enhancement of the MoS_2_ flakes on nanostructures can be attributed to the increased electric field at the excitation and emission wavelengths [2,14]. The comparison of the electric field at the exciton resonance wavelengths and the excitation wavelength (530 nm) clearly suggests that the broadband enhancement of the electric field enables the increased PL and Raman intensities from MoS_2_/Au-NP.

In MoS_2_/Au-NP, the suspended MoS_2_ flakes are above the Au thin film with a 50-nm gap. The suspended MoS_2_ region can absorb reflected light from the underlying Au thin films, raising *A*_MoS2_. As a means of enhanced absorption, insulating spacer layers, such as Al_2_O_3_, HfO_2_, TiO_2_, and SiO_2_, can be inserted between the 2D TMD layers and metal back-reflectors [15,16]. When the Fabry–Pérot resonance condition is met, a large *E* appears on the MoS_2_ flake and increases the *A*_MoS2_ [15]. FDTD calculations showed that the optical reflectance of MoS_2_ monolayers separated by a 50-nm gap above SiO_2_/Si wafers was smaller than that of the MoS_2_/Au-F (Appendix A). This result suggests that the gap in the Au-NP array also contributed to the enhanced optical absorption.

The surface morphology (top) and the line profile of topography (bottom) shown in Figure 6a indicate that the MoS_2_ monolayer flakes transferred onto the Au-NP array had very flat surfaces. This result suggests that the nonstrained MoS_2_ monolayers were suspended between neighboring Au-NPs. Since the surface morphology of the MoS_2_ flakes did not undergo severe deformation, it was very difficult to distinguish the MoS_2_/Au-NP contact region from the suspended region of the surface morphology (Figure 6a). As shown in Figure 1c, the evaporated Au thin films had a rough surface, resulting from migration and agglomeration during growth [15]. In Figure 6a, characteristic features of the agglomerated Au surface can be observed on the MoS_2_ surface, allowing us to identify the MoS_2_/Au-NP contact region (yellow dashed line circles). Figure 6b shows the maps and histograms of the measured CPD in MoS_2_/Au-NP, respectively. The average CPD in the MoS_2_/Au-NP contact region and the suspended MoS_2_ region were estimated by fitting the histograms with two Gaussian functions. The estimated CPD_D_ in the MoS_2_/Au-NP contact region was 85 mV larger than that in the suspended regions. As shown in the topographic image (Figure 6a) and the Raman spectra (Figure 2c), the MoS_2_ monolayer flakes were suspended on Au-NPs without strain. Thus, the CPD_D_ modification likely originated from the strong electronic interactions at the interface of MoS_2_ and Au. Overlap of the wavefunctions and the resulting electron redistribution at the interfaces lead to electronic dipoles and increase the CPD_D_ [13]. A larger CPD_D_ corresponds to a larger electric potential, and hence, the electric potential in the MoS_2_/Au-NP contact region is higher than that in the suspended MoS_2_ region.

Nanostructures with high aspect ratios greatly deform the surface morphology of the MoS_2_ flakes on them, and the local strain modifies the electronic structure of the flake [9,17,18,19]. Tensile strain decreases the bandgap energy of TMD monolayers and increases the local density of excitons in the strained region [17,18,19]. Such strain engineering enables researchers to create wavelength-tunable light emitters [17] and quantum emitters [18,19]. Therefore, both strain effects should be used to explain the physical characteristics of TMD/metal hybrid nanostructures. The electronic interaction at the MoS_2_/Au interface increased the CPD of the MoS_2_ surface, as discussed above. However, the local strain imposed by the nanostructures reduced the CPD of the MoS_2_ flake on them [9]. In MoS_2_/Au-NP, the preparation of nonstrained MoS_2_ flakes led to clear CPD contrast, as shown in Figure 6b.

The first dark-state CPD (CPD_D_) maps were obtained after the sample was stored in the dark for at least 1 h, and the CPD under light illumination (CPD_L_) maps was measured while the sample was irradiated with 660-nm-wavelength light. The wavelength of the light source was identical to the A exciton resonance wavelength of the MoS_2_ monolayer (Figure 4b). Under light illumination, the overall CPD_L_ became larger than the CPD_D_, indicating positive surface charging (Figure 6b) [24]. The SPV signals (≡ CPD_L_ − CPD_D_) were estimated to be 31 mV and 28 mV in the MoS_2_/Au-NP contact region and the suspended MoS_2_ region, respectively. This result suggests that the Au-NPs collected the photogenerated electrons from the whole MoS_2_ flake. In contrast, the SPV of MoS_2_/Au-F was only several mV under identical measurement conditions (Appendix A). The much larger SPV signals in MoS_2_/Au-NP were attributed to the significantly enhanced optical adsorption and the efficient collection of the charge carriers from the suspended MoS_2_ regions.

Plausible scenarios to explain all the experimental results are illustrated in Figure 6c. The photogenerated excitons in the MoS_2_/Au-NP contact region can be separated by the aforementioned interfacial electric dipole, and the electrons are readily transferred to Au-NPs. The dipole formation raised the electric potential at the MoS_2_/Au contact, leading to a large potential gradient around Au-NPs. Such a potential difference is clearly shown in the CPD_D_ map in the dark (Figure 6b), which can drive the separation of excitons and the subsequent drift of electrons towards the Au-NPs. Therefore, positive charging occurs at not only MoS_2_/Au-NP contacts but also in the suspended MoS_2_ regions. After the light source was turned off, the measured CPD_D_ returned to the value of the initial dark state (Appendix A). This suggests that either desorption or chemical reaction on the sample surface cannot explain the measured CPD changes. In Figure 6c, radiative recombination processes are not included for the sake of simplicity.

## 4. Conclusions

We fabricated hybrid nanostructures consisting of MoS_2_ monolayers and periodic Au-NP arrays and investigated their optical and electrical characteristics. Careful transfer processes were used to produce very flat and nonstrained MoS_2_ monolayers on the 50-nm-high Au-NPs. The Raman and PL intensities of the MoS_2_ monolayers on Au-NPs were larger than those on flat Au thin films, indicating enhanced optical absorption of the MoS_2_ monolayers (*A*_MoS2_). The FDTD calculations showed that SPP excitation in Au-NPs enabled broadband enhancement of *A*_MoS2_. In particular, *A*_MoS2_ reached 60% at *λ* = 615 nm, close to the exciton resonance wavelengths of MoS_2_. The CPD maps showed that the surface potential in the MoS_2_ region in contact with the Au-NP top surface was higher than that in the suspended MoS_2_ region due to the strong electronic interaction at the MoS_2_/Au interface. Such potential modulation helped the separation of excitons and the collection of photogenerated charge carriers. The SPV mappings revealed that uniform positive charging occurred on the suspended MoS_2_ surface under illumination of 660-nm-wavelength light. This result suggested that Au-NPs, in addition to the absorption-boosting plasmonic nanostructures, could be used as efficient carrier-collecting electrodes.

## Figures and Tables

**Figure 1 nanomaterials-12-01567-f001:**
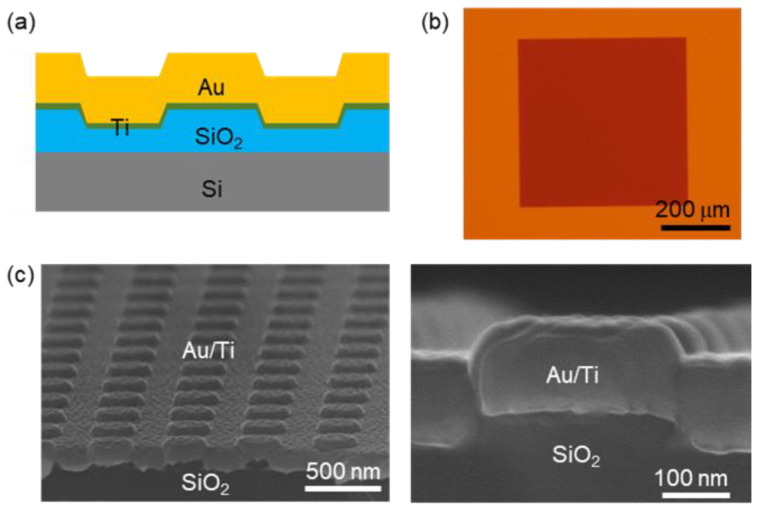
(**a**) Cross-sectional schematic illustration of the Au-NP array. Au (100 nm)/Ti (10 nm) thin films were evaporated on 50-nm-high SiO_2_ NP arrays. (**b**) Top-view optical micrograph of the Au-NP pattern (area: 500 × 500 μm^2^). (**c**) Scanning electron microscope images of Au-NPs obtained at low (left) and high (right) magnifications.

**Figure 2 nanomaterials-12-01567-f002:**
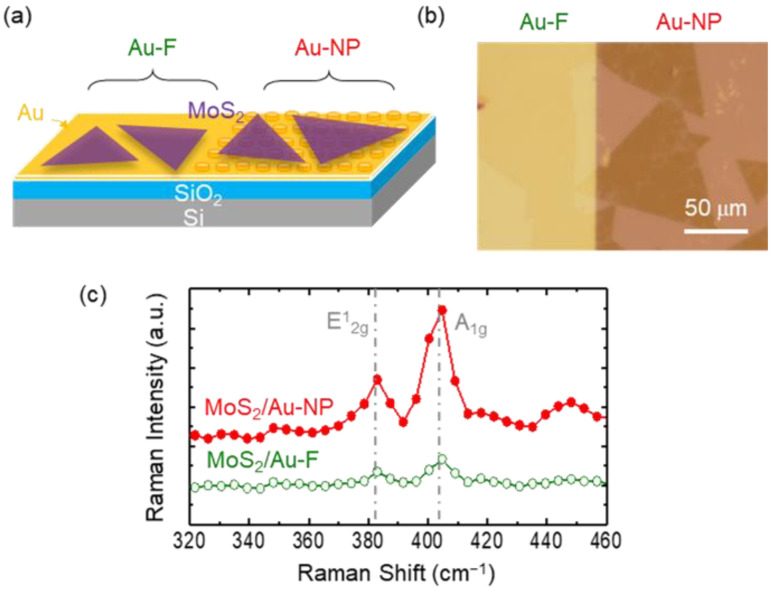
(**a**) Schematic illustration, (**b**) optical micrograph, and (**c**) micro Raman spectra of the MoS_2_ flakes transferred onto the Au-F and Au-NPs.

**Figure 3 nanomaterials-12-01567-f003:**
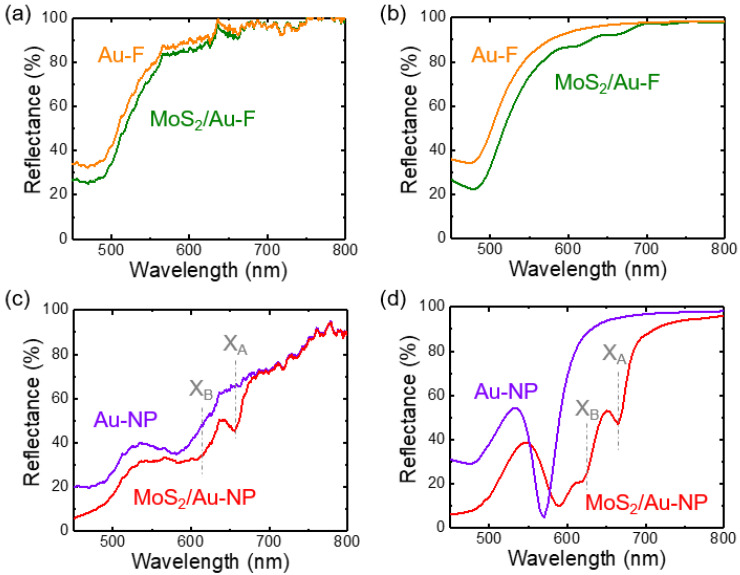
(**a**) Measured and (**b**) calculated reflectance spectra of Au-F (orange) and MoS_2_/Au-F (green), respectively. (**c**) Measured and (**d**) calculated reflectance spectra of Au-NP (violet) and MoS_2_/Au-NP (red), respectively. X_A_ and X_B_ indicate the A and B exciton wavelengths of MoS_2_ monolayers, respectively.

**Figure 4 nanomaterials-12-01567-f004:**
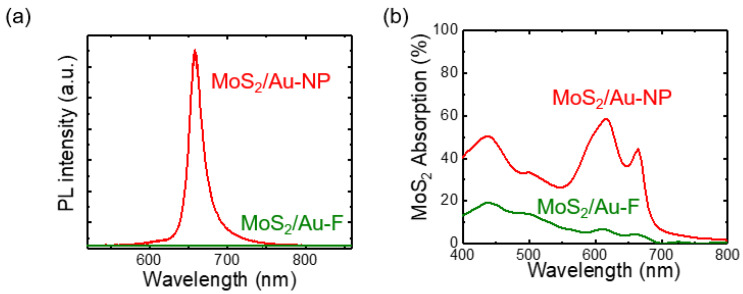
(**a**) Micro PL and (**b**) calculated absorption spectra of the MoS_2_ monolayers on Au-F (green) and Au-NP (red), respectively.

**Figure 5 nanomaterials-12-01567-f005:**
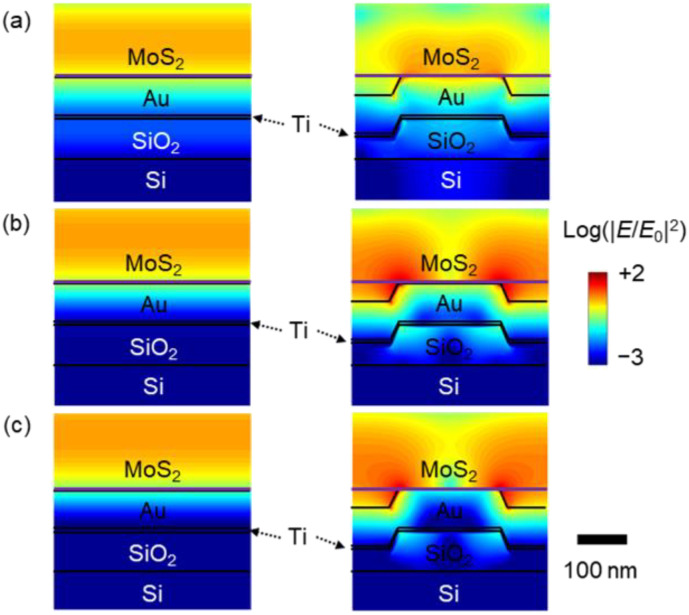
FDTD-calculated electric field (*E*) intensity distribution of MoS_2_/Au and MoS_2_/Au-NP at wavelengths of (**a**) 530 nm, (**b**) 615 nm and (**c**) 660 nm. E_0_ indicates the magnitude of the electric field of the incident light.

**Figure 6 nanomaterials-12-01567-f006:**
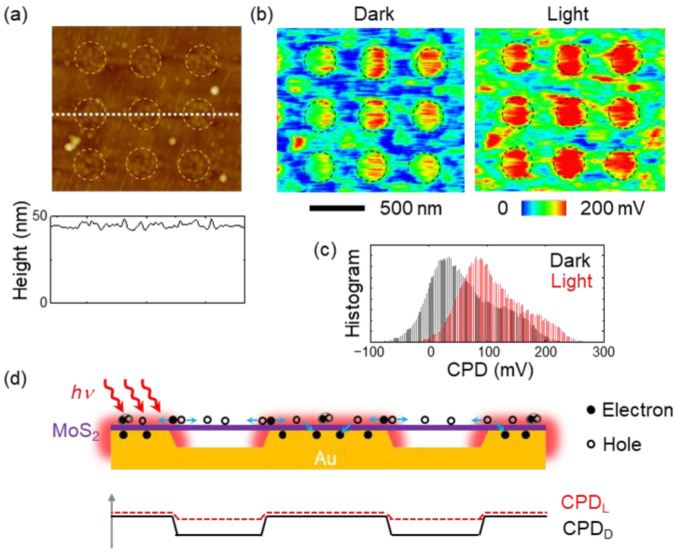
(**a**) Surface morphology map (top) and height profile (bottom) of the MoS_2_/Au-NPs. The profile was obtained along the white dashed line in the map. (**b**) CPD maps of the MoS_2_/Au-NPs. The left and right CPD maps were taken in the dark and under illumination of 660-nm light (power density: 1.6 mW/cm^2^), respectively. The dashed line circles in a and b indicate the regions where the MoS_2_ monolayers were in contact with the top surface of the Au-NPs. (**c**) A histogram of the measured CPD in the maps in b (black: in dark and red: under light illumination). (**d**) Schematic diagrams illustrating the behaviors of photogenerated excitons and charge carriers in MoS_2_/Au-NP. Electron transfer from MoS_2_ to Au leaves holes, leading to positive charging.

## Data Availability

The data presented in this study are available on request from the corresponding author.

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
