# Peer review of "Enhanced Light Absorption and Efficient Carrier Collection in MoS2 Monolayers on Au Nanopillars"

_nanomaterials, 2022, doi:10.3390/nano12091567_

Round 1

Reviewer 1 Report

The authors have fabricated hybrid nanostructures composited by MoS2 monolayers and Au-NP arrays for the enhancement of light absorption and efficient carrier collection. The authors have also demonstrated that by finite-difference time-domain calculations, the Au-NPs have significantly concentrated the incident light near their surfaces to result in the broadband absorption enhancement in the MoS2 flakes. Overall, this work can inspire more material design ideas of MoS2-based materials to efficiently collect photogenerated electrons. Therefore, I would like to recommend this work to publish in Nanomaterials. Below are a few suggestions for the authors.

1. For Figure 1c, the authors should separately provide the description of these two SEM image in the caption.

2. Photoluminescence of pure MoS2 flakes should be provided in the supporting information to reveal their optical property.

3. For the introduction “Two-dimensional (2D) layered transition metal dichalcogenides (TMDs) are emerging as candidates for novel electronic...”, more references could be cited to broaden the introduction.

https://doi.org/10.1021/acssuschemeng.1c01868

Reviewer 2 Report

In this paper the authors present investigations on MoS2 deposited, as a planar single layer film, on top of a plasmonic pillars array.

The enhanced raman and PL are presented to support the expected enhanced factor due to the plasmonic structures. Then  the platform is use to collect photocurrent with also some enhancement due to the direct contact with the Au layer at the pillar / MoS2 interface

The manuscript is well written and the results sound correct, anyway, the use of MoS2 over plasmonic nanostructures is not new at all and a quite large number of papers exist on that topic. 

I think that first of all the authors must improve their introduction on the state-of-the-art trying to better underline some novelty aspects reported here. The deposition of MoS2 without curved area is interesting although not significant for the final application.

Minor comments on the paper are:

  1. fig4a reports the PL, it is strange to me that no signal can be seen from MoS2 on flat Au
  2. Line 193. the authors say "strong concentrate". strong is rather an overstatement, this plasmonic platform cannot "strongly" concentrate anything. the enhancemend factor and the field confinement is very low
  3. fig 5. How the low enhancement factor can justify the good enhancement in PL and SERS?

Some recent literature on the topic (on very similar papers) to be included are:

doi.org/10.1016/j.optlastec.2020.106306

doi.org/10.1021/acsami.1c21960

DOI: 10.1039/D0NR08158B
